# Impact of Physical Activity Counselling on Children with Medical Conditions and Disabilities and Their Families

**DOI:** 10.3390/children10081293

**Published:** 2023-07-27

**Authors:** Hannah C. Cummings, Jordan Merkas, Jenna Yaraskavitch, Patricia E. Longmuir

**Affiliations:** 1Faculty of Medicine, University of Ottawa, Ottawa, ON K1H 8M5, Canada; hcumm012@uottawa.ca (H.C.C.); jmerk016@uottawa.ca (J.M.); 2Children’s Hospital of Eastern Ontario Research Institute, Ottawa, ON K1H 8L1, Canada; jyaraskavitch@cheo.on.ca; 3Faculties of Medicine and Health Sciences, University of Ottawa, Ottawa, ON K1H 8M5, Canada

**Keywords:** behaviour therapy, child, counselling, exercise, health promotion, patient satisfaction

## Abstract

Physical activity counselling can target cognitive-affective participation barriers, but counselling benefits for children with medical conditions/disabilities were unknown. This study investigated successes, challenges, and the impact of physical activity counselling on children and their families. One-on-one semi-structured interviews were completed with 7 patients (2 male/5 female, aged 13–17) and 4 parents who participated in 2–8 weekly counselling sessions (2015–2020). Interviews were recorded and transcribed verbatim for inductive thematic analyses. Counselling encouraged positive mindset changes (viewing physical activity more holistically, making it “more fun and manageable”, helping them to “learn how to love moving and doing sports”). Participants felt strong support (feeling heard, validated, and provided with “hope… that we can still achieve things… even though it may seem like there’s limitations”). Counselling was viewed positively. The intent to improve active lifestyle attitudes and confidence was reflected in positive, primarily cognitive-affective (motivation for activity, “more general skills of having a positive attitude towards physical activity and the willingness to try new things”) outcomes. More sessions, additional resources to keep, and follow-up after counselling completion were recommended to support behaviour change. Future research should evaluate enhanced counselling services and comparing children who have and have not received such counselling.

## 1. Introduction

Physical activity positively impacts the physical, emotional, and social development of children [1]. The World Health Organization recommends that children perform 60 min of moderate-to-vigorous intensity activity and 2–3 h of light intensity activity daily, with bone and muscle strengthening activities 2–3 days per week [2]. Although physical activity is crucial for the health of children living with chronic medical conditions or disabilities, and proves beneficial to health regardless of diagnosis [3], many barriers to physical activity persist for these children. As a result, children with medical conditions or disabilities are much less likely to engage in healthy, active lifestyles [1,4,5,6]. Active lifestyles provide a variety of physical and mental health benefits, such as improved cardiovascular health, decreased anxiety and depression, and decreased obesity risk [7,8]. The more sedentary lifestyles of children with medical conditions and disabilities negatively impacts their socialization with peers, growth and development, and academic readiness [9] and increases their risk for secondary morbidities throughout life [7].

The Children’s Hospital of Eastern Ontario (CHEO) is a tertiary paediatric hospital that uniquely offers physical activity counselling as part of clinical care to children with any type of medical condition or disability. While physical activity is broadly recommended for these children [10], reports of providing such supports are limited to programs that treat childhood obesity [11]. Clinicians refer patients to this program in order to equip them and their families with the tools needed to achieve healthy physical activity levels. The novel one-on-one program offered at CHEO analyzes the child’s current goals and activity levels to tailor a physical activity plan and set specific and attainable individualized goals with the child. 

Given the unique nature of the program, little information exists about the participants’ and families’ experience of receiving such support. The aim of this study was to evaluate the provision of physical activity support as part of clinical care from the child and parent perspective. The study also sought to identify the program’s strengths and understand participant-identified areas for improvement. The goal was to understand how to optimize the physical activity counselling program to meet the requirements of paediatric patients and their families, with the long-term goal of enhanced and longer-lasting improvements to the children’s physical activity participation.

## 2. Materials and Methods

This study was a cross-sectional analysis of former participants in CHEO’s Physical Activity Counselling program (CHEO Research Ethics Board approval 21-66X). All participants provided verbal informed consent prior to study participation. In accordance with the research ethics approval, study data are available by request to the corresponding author. 

### 2.1. Participants

The study population included the 72 children who had participated in CHEO’s Physical Activity Counselling program between January 2015 and December 2020. Participants eligible for this study were aged eight or older, regardless of the child’s sex/gender or the divisions/departments that treated the patient. The Research Ethics Board did not permit contact with children no longer followed at CHEO. The parents or legal guardians of counselled children were also eligible to participate in this study if they were involved or supported the physical activity counselling. Parents or guardians of children excluded from participating were not interviewed.

After screening for eligibility, the referring providers for all eligible participants were contacted regarding participant recruitment. If the referring provider was no longer treating the participant, their most recent provider at CHEO was contacted regarding study recruitment. For those providers that accepted, they signed a recruitment letter that was then mailed out to their patients. Two weeks later, participants were contacted via telephone and were given the option to participate or decline. Parents were asked if they were involved with or supported their child’s counselling and, if so, were given the opportunity to participate.

### 2.2. Design

The study required one study session that consisted of a one-on-one semi-structured interview. Given the COVID-19 pandemic, interviews were conducted online via Zoom or by telephone. Interviews were audio-recorded and transcribed verbatim and notes were taken to enable coding and inductive thematic analyses assisted by NVivo software (QSR International, release 1.6.1, Burlington, Massachusetts, United States). Children and parents/legal guardians were to be interviewed separately. If the child was uncomfortable being interviewed alone, they were interviewed with the parents/legal guardians present.

### 2.3. Outcome Assessment

The one-on-one interviews were semi-structured using a discussion guide that identified topics and questions of interest, with the direction and responses to these topics being guided by the participants. Table 1 lists the questions and prompts utilised to interview child participants. Table 2 provides details of the questions and prompts utilised for parent interviews. The complete interview guides are provided in the Appendix A. The interviews sought input on what participants remembered, what was helpful, what could be improved, if and how it changed their behaviour, if they would recommend it to others, and whether they would repeat the program. During the parent/legal guardian interviews, particular emphasis was on the family perspective and the triangulation of data between child and parent.

### 2.4. Intervention Delivery

Counsellors, supervised by registered kinesiologists, are students in the Physical Activity and Intervention Counselling Master of Human Kinetics program at the University of Ottawa. The counsellors receive year-long training where they develop and practice a variety of therapeutic approaches, including but not limited to person-centered therapy, cognitive behavioural therapy, and gestalt therapy. One of the approaches involved in person-centered therapy is motivational interviewing [12], which is associated with behaviour change and physical activity maintenance [13]. The therapeutic approaches are supported by counselling skills, including active listening, reflection, attending behaviour, empathy, observation and questions to foster a positive and productive relationship with patients [14].

During the initial sessions, counselors evaluate patient’s readiness for change and discuss appropriate expectations of the program. Given the breadth of patient diagnoses and readiness for change, tailored counseling is required. Session length and content differs, depending on the direction/goals of the patient and availability of the counselors. Counselling occurs at CHEO, in the community, or virtually and typically consists of one session per week for six to eight weeks. 

Upon referral, counsellors work to equip the child with the tools needed to achieve healthy physical activity levels. Counselling involves a variety of activities, using self-regulation techniques such as self-monitoring, motivation and self-awareness targeting, creating personalized activity plans, and building activity confidence. Goal setting is conducted using the SMART-EST principles (Specific, Measurable, Attainable, Relevant, Time-bound, Evidence-based, Strategic, and Tailored) to ensure individualized and appropriate goals are developed [15]. Counsellors assist in identifying barriers to adopting the activity plan and address barriers with patients.

### 2.5. Data Analysis

All recorded interviews were transcribed verbatim, with the audio recordings retained and used to supplement the written transcripts during analyses. Inductive thematic analyses were independently conducted by two researchers (HCC, JM) using the model of Braun and Clarke [16]. Each researcher familiarized themselves with the data by reading the transcripts and listening to the audio recordings. Key concepts were identified inductively as they emerged from the data, generating initial codes. Member-checking was then performed, with participants reviewing the initial codes and their interview transcripts/notes by phone or email. The research team reviewed the key concepts and resolved any discrepancies through discussion. The concepts were grouped into themes by the principal investigator (HCC), guided by a pre-established coding framework (perceptions, benefits, desired changes), then presented to the research team for discussion prior to finalizing the defining themes. Disagreements were resolved through consensus of the research team.

## 3. Results

### 3.1. Participant Recruitment and Demographics

Of the 72 children who had been referred for physical activity counselling as part of their clinical care, 7 children and four parents completed a research study interview (Figure 1). Children were primarily excluded because they were no longer followed at CHEO (21/44), they did not complete the physical activity counselling after the referral (9/44), or because they received group rather than individual counselling (6/44). Most eligible children (25/28) were contacted by the research team regarding participation in this study. Of the 18 children who declined to participate, 5 indicated that they did not have anything to say and 10 did not provide a reason. Parents were primarily excluded because they were not involved with the physical activity counselling. The four parents who did participate in the counselling agreed to be interviewed.

Nine families were represented among the study participants. The seven children who agreed to participate were 13 to 17 years of age at the time of being interviewed (Table 3). The participants had a diverse range of medical conditions and disabilities, reflecting the provision of physical activity counselling to all pediatric patients. Interviews were completed by two parents of participating children and two additional parents whose children were not interviewed.

### 3.2. Positive Mindset Changes

Counselling helped participants view physical activity with a more holistic and positive mindset, challenging their negative preconceived notions. Prior to counselling, physical activity “was really something you had to do, impossible to have fun doing it, impossible to have fun doing sports” (C2). Tailoring activities to fit each participant’s interests and abilities facilitated reframing their mindset towards being active. 

“…when somebody else… opens another direction for you, it’s a lot easier to go in that direction, versus opening it by yourself.” (C5)

Counselling “made physical activity more fun and more manageable” (P1). Counsellors did this by proposing “different activities [participants] could do to learn how to love moving and doing sports” (C2). Parents also noted their child was more open-minded and positive towards physical activity (PI). For some, this change in perspective was sustained long after completion of the counselling program.

### 3.3. Strong Support by Counsellors

Ten of eleven participants felt well-supported in their counselling. In a chronic illness context, support can be very meaningful. 

“When you have a complex child who can’t walk, talk, or eat, having someone come in who is kind, listens, will work with what they can, come week after week, do the best they can, are pleasant, and share experiences… It affects you emotionally on a higher level. It doesn’t matter what they do, but someone cares enough to try. No one else can get it. It affects you at a higher level.” (P3)

Support extended beyond the physical aspect as participants also felt emotionally supported. Parents particularly mentioned feeling a sense of validation and renewed hope. One parent mentioned their counsellor provided a strong sense of hope, stating that “there are nice people that still care” (P3). Another parent agreed their counsellor empowered them, despite their child’s limitations: “…it was the hope and the support that we can still achieve things, and even though it may seem like there’s limitations, really there are workarounds to them” (P1).

Participants particularly enjoyed feeling heard by their counsellor and appreciated the space to speak their mind: “it’s just… nice to be heard. And… to be able to say that you feel in your heart you are understood” (C2). The unique relationship with their counsellor provided a resource in which to confide and seek support, beyond the family. 

### 3.4. Long-Term Impact

Many participants had difficulty remembering the details of counselling, either due to length of time since the intervention, “it’s just because a long… year gap in between. So I don’t…really remember anything” (C3); or lack of impression, “...I don’t frankly think I was taught anything. And if I was, I’m just, I don’t remember at all” (C7). Although most found the program beneficial while ongoing, many either briefly continued the learned behaviours and lost them over time or stopped altogether when the counselling sessions ended. One even mentioned that, “until I got the letter inviting me to possibly join the study, like I’d forgotten that I’d done [the physical activity counselling] at all” (C4).

Despite lack of long-term impact in behaviour change noted by participants, many of their responses demonstrated an open, positive mindset towards physical activity. It appears much of the long-term change occurred at the cognitive-affective level rather than the physical level. 

“I think some of the more general skills of having a positive attitude towards physical activity and the willingness to try new things, which is really challenging to his medical conditions, it’s definitely more present than it was before the counselling.” (P1)

The majority of participants agreed they would repeat the program if given the opportunity. Many would modify their approach to the program and use strategies to help maximize their experience and better retain what they learned. 

### 3.5. Suggestions for Improvement

Participants and their families had excellent suggestions for improving the physical activity counselling based on their unique experiences. Although some had their experience cut short due to the COVID-19 pandemic, even those who completed the program would increase the length of counselling. One mother said “there could’ve been more sustainable output. If there were… more [sessions]” (P1). 

Children mentioned that resources/handouts would be helpful to integrate the concepts into their routine after program completion. One suggested electronic or physical documents to help them remember (C3). Another who kept notes of her experience stated that having those records was a big part of her retention (C6). 

Implementing follow-up sessions after completion of the initial counselling program was a common suggestion. Follow-ups would serve as helpful reminders and build motivation to continue the behaviours introduced during counselling. 

## 4. Discussion

### 4.1. Benefits of Physical Activity Counselling

The importance of enabling children to be provided with support for physically active lifestyles through the healthcare system has been recognized for many pediatric clinical populations [17,18,19]. The role of pediatricians is critically important in that they can guide patients and families towards meeting current guidelines, facilitate physical activity and physical literacy assessments, and advocate for activity opportunities in schools and communities [17]. The availability of physical activity support is particularly important for children with special healthcare needs, as their active lifestyle participation rates are even lower than peers [17]. In this study, physical activity counselling helped participants view physical activity from a positive and holistic perspective. Participants felt well-supported by counsellors, creating feelings of validation and hope. These results support calls for enhanced physical activity support in pediatric clinical populations. Specialized programs, extending beyond the physician’s scope, are needed to support healthy, active lifestyles in these children [17].

### 4.2. Need for an Individualised Approach

The main strength of CHEO’s physical activity counselling program was the child-counsellor relationship. Most of the participants were referred for support to increase physical activity motivation and the importance of a healthy lifestyle (Table 3). In these initial stages of behaviour change, the focus of counselling is on helping the patient to understand potential benefits, address the pros and cons of change, and examine perceived barriers to change [20]. To be effective, each of these counselling activities requires a positive and supportive relationship between the patient and counsellor and the ability to focus on the unique perspectives of each individual patient. The importance of the counselling relationship has previously been identified for children with cystic fibrosis [19] and among adults and children with disabilities [20]. Research among adults has also indicated their strong support for a customized approach and the importance of the participant-counsellor relationship [21]. Children and parents in this study emphasized that they believe they would obtain even greater benefits if the number of sessions was increased and if follow-up appointments or handouts for future reference were provided. These findings align with current physical activity counselling knowledge among adults, who have also expressed the need for longer-term support and follow-up after program completion to realize greater benefits [21].

### 4.3. Targeting Physical Activity Perceptions and Motivation

The primary goal of most physical activity counselling sessions was to develop positive attitudes towards physical activity. Although sessions often incorporated changes to behaviour (e.g., walking or playing games during the session), the goal was to have patients learn activities they could do successfully and would enjoy. Children reported that the counselling changed their view of physical activity from “something you had to do” (C2) to something “more fun and more manageable” (P1). The more positive mindset reported by participants, which was sustained long-term, reflects this focus on changing activity perceptions.

Behaviour change theory offers insight into why counselling had a greater impact on behaviour in the short versus long-term. Many environmental (transportation, availability), psychosocial (discomfort with physical activity, social influences), and personal (lack of energy, motivation, medical condition) factors are important when adopting new behaviours [22,23,24]. Taken together, it is not surprising that long-term behaviour changes would be challenging with only six to eight physical activity counselling sessions. Research suggests there is tremendous individual variation in the timeline for establishing new health behaviours. Study participants required 18 to 254 days to firmly establish a new habit, with most participants requiring an average of 66 days [25].

Readiness for change is also important when trying to change behaviour [22,26]. In each readiness stage, the child’s needs are different. The pre-contemplation stage is when the child lacks information and understanding regarding the need for behaviour change [27]. At this stage, directing the child to engage in physical activity would not help them understand the rationale for behaviour change or provide them with the motivation to participate. Understanding and addressing the child’s perceptions, barriers and reasons for ambivalence help to enhance their readiness for behaviour change [26,28]. Most participants in this study were referred in the pre-contemplation or contemplation stage, which aligned with the counselling goal to build physical activity motivation, confidence, and knowledge as well as address activity ambivalence [27]. Participants reported long-term changes in their mindset indicating that the initial counselling had moved them to a position in which they would be more prepared to act. The desire for more counselling sessions may have reflected their increased readiness for behaviour change and their progression into the preparation (planning to act) or action stages of change [27]. However, the counselling and support strategies needed to move someone from a stage of readiness to action differ from those that focus on motivation for activity. Increasing the number of counselling sessions would provide additional opportunities for children who initially are not motivated to be active to move through the behaviour change stages and engage in the goal setting, self-assessment and relapse prevention activities needed to support and sustain a healthy, active lifestyle [20].

### 4.4. Strengths and Limitations

This initial study examining child and parent perceptions of receiving physical activity counselling support as part of their clinical care provides novel and important information for clinicians and families. Additional research with a larger sample of participants would enable more extensive analyses by demographic variables (e.g., sex, age, time since counselling). Results from this initial study should be interpreted in light of the identified strengths and weaknesses discussed below.

The diagnostic diversity of study participants was a strength of the study as it provided the opportunity to obtain perspectives from children facing a wide variety of participation barriers, which broadens the applicability of the study results. Children excluded had cardiology, developmental, neurology, obesity, or unknown conditions. Those approached to participate had cardiology, developmental, neurology, concussion, mental health, obesity, or unknown conditions. Therefore, the diagnoses were similar between those excluded and approached to participate, except that children with concussion or mental health diagnoses were not excluded. The diagnoses represented among participants approached for the study were similar, regardless of whether or not the child agreed to participate.

Qualitative research typically relies on a small, purposively selected sample of participants in order to support a more in-depth analysis of each identified case [29]. The sample size in this study was smaller than originally anticipated due to the large proportion of patients (61%) excluded from participation. Most were excluded because they were no longer followed at CHEO (48%; research ethics board requirement), they participated in group rather than individual counselling (14%), or they did not complete the counselling after referral (20%). Among those eligible to be approached, seven children and four parents from nine families agreed to participate (9/28 (32%) recruitment rate). The 9 participating families were very similar to the minimum sample size of 10 that is recommended for studies, such as this, that focus on a limited scope of inquiry [30].

Two of seven child participants were male, making it difficult to assess the impact of sex. Both male participants were young when counselled three to four years prior, impacting program recall. It was also not possible to determine the impact of recall bias on the study results. It is possible that participants more easily recalled positive or negative (rather than neutral) aspects of the counselling experience. Interviews were led by a researcher (HCC) who was not connected to the physical activity counselling service or the clinical care of participants in order to encourage honest interview responses. Nevertheless, the study’s conclusions regarding the long-term effects of physical activity counselling may have been impacted by recall difficulties.

## 5. Conclusions

This study was novel in its examination of a physical activity program uniquely provided as part of clinical care for children with any type of medical condition or disability. Interviews with previous counselling participants highlighted the importance of the child-counsellor relationship and the impact of the counselling sessions on child and parent perceptions of physical activity. Physical activity counselling was viewed positively by children and parents, regardless of the child’s diagnoses or reason for referral. Counselling was designed to build, and effectively built, long-term motivation, enjoyment, and confidence for physical activity. Participants also benefitted by learning more about the physical activity options available, given the limitations of each child’s medical condition. Feedback unanimously found that having more counselling sessions, resources to keep, and follow-up after the end of the sessions could help participants remember and continue to implement what was learned. Results of this study support the need for expanded access to physical activity counselling support in clinical care, services that are currently offered only through obesity treatment programs. Counselling services must be tailored to each child’s interests, abilities, medical condition and readiness for behaviour change. The six to eight counselling sessions provided in this study were sufficient to change physical activity perceptions and enhance physical activity motivation over the long-term. Future research should examine participant perceptions after implementation of the changes recommended through this study. Such research should be conducted both immediately and longer-term following the end of counselling. Examining physical activity attitudes and participation over the long-term among children who have and have not received such counselling is also recommended. Future efforts should also investigate the additional and different counseling support that would be required to effectively target changes to activity behaviour.

## Figures and Tables

**Figure 1 children-10-01293-f001:**
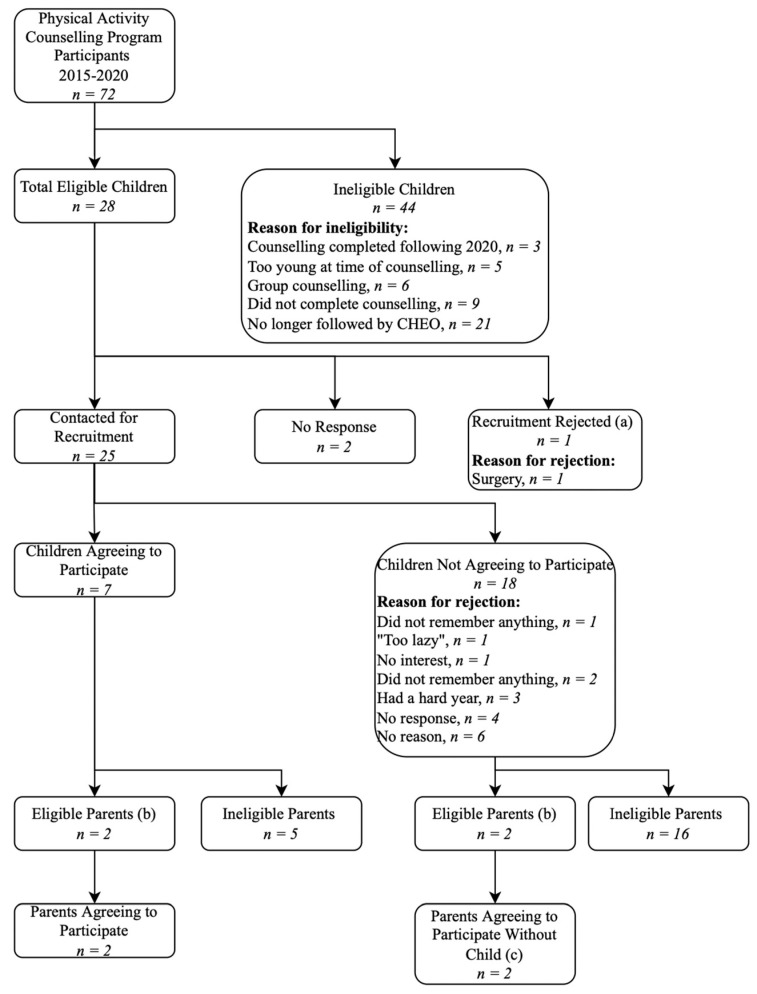
Flowchart demonstrating the recruitment process and inclusion/exclusion of potential participants. (a) Most responsible physicians did not approve patient recruitment. (b) Eligible parents were involved in or supported their child’s counselling sessions. The remaining parents/guardians did not meet this criterion. (c) Parents were able to participate even if their child was unwilling or unable to participate.

**Table 1 children-10-01293-t001:** Semi-structured interview questions for children.

Question	Prompts
What do you remember from physical activity counselling?	Why do you think you remember that the best? Do you still use anything that you learned through physical activity counselling? How often do you think about or apply the things you learned in physical activity counselling?
How did the counselling change your physical activity?	How active are you today? Is your activity level today impacted by your participation in physical activity counselling? What did physical activity counselling not impact? Do you think that physical activity counselling made a difference in your life today? Why or why not?
What was helpful?	What did you learn from participating in physical activity counselling? What did you enjoy about the experience? Did you feel supported throughout the process? Why or why not? Is there anything you really enjoyed?
What could be improved?	What do you think could be added to support you better? What was not helpful? Is there anything you really did not enjoy? Would you change the format? What do you think would have made a bigger impact?
Would you do it again?	Why would you like to do it again? Would you do anything differently if you were to do it a second time?
Would you recommend that other children or families do it?	Why do you think other children should do it? Should we do anything differently if other children were to do it?

**Table 2 children-10-01293-t002:** Semi-structured interview questions for parents.

Question	Prompts
What do you remember from your child’s physical activity counselling?	Why do you think you remember that the best? Does your child still use anything that they learned through physical activity counselling? How often do you think they think about or apply the things they learned in physical activity counselling?
How did the counselling change your child’s physical activity?	How active is your child today? Is your child’s activity level today impacted by their participation in physical activity counselling? What did physical activity counselling not impact? Do you think that physical activity counselling made a difference in your child’s life today? Why or why not?
How did the counselling change your understanding of your child’s physical activity?	Are you more confident about the activities appropriate for your child? Do you think you are better able to support and encourage your child’s physical activity?
What was helpful?	What do you think your child took away from the experience? What were your main takeaways from participating in physical activity counselling? What do you think your child enjoyed about the experience? What did you enjoy about the experience as a parent? Do you think your child felt supported throughout the process? Why or why not? Did you feel supported throughout the process? Why or why not? Is there anything your child really enjoyed? Is there anything you really enjoyed?
What could be improved?	What do you think could be added to support your child better? What do you think could be added to support you better? What was not helpful for your child? What was not helpful for you? Is there anything your child really did not enjoy? Is there anything you really did not enjoy? Would you change the format? What do you think would have made a bigger impact for your child? What do you think would have made a bigger impact for yourself?
Would you do it again?	Why would you like to do it again? Would you do anything differently if you were to do it a second time?
Would you recommend that other children or families do it?	Why do you think other children/families should do it? Should we do anything differently if other children/families were to do it?

**Table 3 children-10-01293-t003:** Demographics for those participating in this study.

ID	Child or Parent	Child’s Age at Interview	Age at Counselling	Child’s Sex	Child’s Diagnosis	Reason for Referral	# of Sessions	Comments
C1	Child	13	10	Male	Epilepsy	Encourage healthy active lifestyle	8	C1
C2	Child	14	8	Female	Epilepsy, obesity	Healthier lifestyle	8	Interview conducted in French
C3	Child	17	13	Male	Congenital heart defect	Not indicated	2	
C4	Child	16	14	Female	Asperger’s	Increase physical activity, use activity to connect with peers, anxiety management	5	Parent in room during interview
C5	Child	16	14	Female	Concussion	Anxiety with return to play	3	
C6	Child	17	14	Female	Concussion	Return to play	3	
C7	Child	16	13	Female	Cystic fibrosis	Physical activity counselling	6	
P1	Parent	13	10	Male	Epilepsy	Encourage healthy active lifestyle	8	Parent of C1
P2	Parent	14	8	Female	Epilepsy, obesity	Healthier lifestyle	8	Parent of C2; interview conducted in French
P3	Parent	19	16	Male	Global developmental delay	Individual plan for physical activity/exercise	6	Child unable to participate due to being non-verbal
P4	Parent	18	16	Male	22q11.2 deletion, congenital heart defect	Improve fitness and endurance	2	Child unwilling to participate, no interest

## Data Availability

The data supporting the study findings are available from the corresponding author, PEL, upon request.

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
