# Peer review of "Impact of Physical Activity Counselling on Children with Medical Conditions and Disabilities and Their Families"

_children, 2023, doi:10.3390/children10081293_

Round 1

Reviewer 1 Report

Reviewer Report:

Title:

The title "Impact of physical activity counselling on children with medical conditions and disabilities and their families" accurately reflects the content of the manuscript. It provides a clear overview of the study's focus on examining the effects of physical activity counselling on children and their families.

Positive Points:

1. The abstract provides a concise summary of the study, including the objectives, methods, and key findings. It effectively conveys the main points of the research.

2. The introduction provides relevant background information on the importance of physical activity for children with medical conditions and disabilities. It establishes the need for physical activity counselling programs and highlights the unique nature of the program being studied.

3. The methods section provides a clear description of the study design, participant recruitment process, and data analysis methods. It offers sufficient detail for replication and establishes the credibility of the research.

4. The results section presents the main findings of the study in a clear and organized manner. The use of participant quotes adds depth and authenticity to the results.

5. The discussion section provides a comprehensive analysis and interpretation of the findings. It discusses the strengths of the physical activity counselling program and highlights areas for improvement. The section also relates the findings to existing literature and suggests avenues for future research.

Areas to be improved:

1. One negative aspect is the limited sample size and lack of diversity among participants. The study only included 7 children and 4 parents, making it difficult to generalize the findings. Additionally, the gender imbalance (2 male participants) restricts the ability to assess the impact of sex on the outcomes. The authors should acknowledge these limitations and discuss potential implications for the generalizability of the findings.

2. Another limitation is the recall bias mentioned by some participants, particularly regarding the long-term impact of the counselling program. This should be addressed, and the authors should acknowledge that the study's conclusions regarding long-term effects may be limited due to the difficulties in recall.

3. While the study provides valuable insights into the positive impact of physical activity counselling, it would benefit from a more detailed description of the specific counselling techniques and strategies employed. This additional information would enhance the clarity and replicability of the study.

Overall, the manuscript provides a valuable contribution to the understanding of the impact of physical activity counselling on children with medical conditions and disabilities. The study design and analysis are rigorous, and the findings are well-presented. However, the limitations regarding sample size, recall bias, and the lack of detailed description of counselling techniques should be addressed. With these revisions, the manuscript will be suitable for publication in its current form.

Nil 

Reviewer 2 Report

In this study, the authors investigated successes, challenges, and the impact of physical activity counselling on children and their families.
This article looks more like a draft. It's short (the introduction and discussions are very short). Although the structure is well organized, statistics cannot be made and pertinent conclusions cannot be drawn, only interviewing 7 participants, you need more participants. The questionnaires should be more elaborated, 6 or 7 questions are too few.
The article presents 12 references. And these are too few.
In my opinion, the article cannot be published in this form. I recommend you to expand the study (in all aspects) and redo it.

Round 2

Reviewer 1 Report

The authors have addressed all the comments raised by me. The manuscript can be accepted in its current form. 

nil

Reviewer 2 Report

The authors have significantly improved the article. They responded correctly and promptly to recommendations.